# LEVERAGING WORD GUESSING GAMES TO ASSESS THE INTELLIGENCE OF LARGE LANGUAGE MODELS

## ABSTRACT

The automatic evaluation of LLM-based agent intelligence is critical in developing advanced LLM-based agents. Although considerable effort has been devoted to developing human-annotated evaluation datasets, such as AlpacaEval, existing techniques are costly, time-consuming, have limited scalability, and lack adaptability. In this paper, inspired by the popular language games "Who is Spy" and "SpyFall", we propose to use the word guessing game to assess the intelligence performance of LLMs. Given a word, the LLM is asked to describe the word and determine its identity (spy or not) based on its and other players' descriptions. Ideally, an advanced agent should possess the ability to accurately describe a given word using an aggressive description while concurrently maximizing confusion in the conservative description, enhancing its participation in the game. To this end, we first develop SpyGPT to evaluate LLMs' expression and disguising abilities. SpyGPT requires the target LLM to describe the given word in aggressive and conservative modes and utilizes the SOTA GPT-4 to determine whether the descriptive sentences can accurately describe the given word. We then introduce SpyGame, an interactive multi-agent framework designed to assess LLMs intelligence through participation in a competitive language-based board game. Incorporating multi-agent interaction, SpyGame requires the target LLM to possess linguistic skills and strategic thinking, providing a more comprehensive evaluation of LLMs' human-like cognitive abilities and adaptability in complex communication situations. The proposed evaluation framework is very easy to implement. We collected words from multiple sources, domains, and languages and used the proposed evaluation framework to conduct experiments. Extensive experiments demonstrate that the proposed SpyGPT and SpyGame effectively evaluate the capabilities of various LLMs, capturing their ability to adapt to novel situations and engage in strategic communication.

## 1 INTRODUCTION

Large language models (LLMs) like ChatGPT, GPT-4 (OpenAI, 2023) and Bard, have recently shown remarkable performance across a wide range of tasks, significantly advancing the field of artificial general intelligence (Bubeck et al., 2023). Concurrently, there has been an increasing focus on developing LLM-based agents for applications in social science (Park et al., 2022; 2023) and engineering (Li et al., 2023a; Qian et al., 2023) domains, with the aim of addressing real-world challenges or social simulation. The most fundamental capabilities required for these LLM-based agent applications are language intelligence and theory of mind intelligence (Kosinski, 2023).

As a result, the automatic evaluation of LLM-based agent intelligence has become crucial for further advancements. The evaluation of LLMs has evolved from focusing on NLP tasks (e.g., GLUE (Wang et al., 2018), MMLU (Hendrycks et al., 2020)) to alignment evaluation (e.g., AlpacaEval (Li et al., 2023b), ) and, ultimately to complex real-world tasks (e.g., Webshop (Yao et al., 2022), Webarena (Zhou et al., 2023)). Generally, these evaluation benchmarks are built on established annotated datasets to assess and analyze the performance of LLMs. However, traditional human-annotated evaluation techniques face challenges such as high costs, time-consuming processes, limited scalability, lack of adaptability, and susceptibility to data leakage.

In contrast to conventional human-annotated evaluation, recent attempts have been to utilize game-playing for assessing the intelligence of LLMs, such as GameEval (Qiao et al., 2023). These approaches aim to provide a more engaging and interactive means of evaluating LLM performance in various tasks and scenarios. In this study, we propose a novel approach to assess the intelligence of LLMs through word guessing games, focusing on two distinct aspects: 1) the ability to accurately describe words for enhancing self-understanding and 2) the ability to intentionally disguise descriptions by being deliberately conservative. These two aspects are related because they evaluate the LLM's capability to generate meaningful and contextually appropriate descriptions. However, they focus on different dimensions of the LLM's intelligence and can be seen as complementary rather than contradictory. The relationship between the two aspects can be seen as a balance between providing information (accurate descriptions) and maintaining intrigue (disguising through conservative descriptions).

In this paper, we propose two frameworks, SpyGPT and SpyGame, to evaluate the capabilities of LLMs in various aspects. SpyGPT, a single-agent direct evaluation method, focuses on assessing LLMs' expression and disguising abilities by requiring the target LLM to describe a given word in both aggressive and conservative modes, while utilizing the state-of-the-art GPT-4 to determine the accuracy of these descriptions. On the other hand, SpyGame is a highly interactive multi-agent framework designed to evaluate LLMs' intelligence through their participation in language-based board game "Who is Spy". By incorporating multi-agent interactions, SpyGame requires the target LLM to exhibit expressive language skills and strategic thinking abilities, thereby providing a more comprehensive assessment of LLMs' human-like cognitive capabilities and adaptability in complex communication situations.

In summary, the contributions of this work are detailed as follows:

- We propose to use word guessing games to assess the intelligence of LLMs. We develop a single-agent framework SpyGPT and a novel interactive multi-agent framework SpyGame, to build a more comprehensive evaluation with a focus on their human-like cognitive abilities and adaptability in complex scenarios.

- We demonstrate the effectiveness of SpyGPT and SpyGame in differentiating the intelligence levels of various LLMs, providing valuable insights for future research and development in the field of artificial general intelligence.

- Experimental results reveal that our proposed frameworks successfully distinguish between the performance of open-source and closed-source LLMs, highlighting the strengths and weaknesses of each model in terms of context comprehension, description accuracy, and the ability to generate ambiguous representations. These findings contribute to a deeper understanding of LLM capabilities and inform the development of more advanced and intelligent language models.

## 2 PRELIMINARY STUDY: SPYGPT

In this section, we present a straightforward and efficient approach, SpyGPT, as a preliminary investigation to examine the capacity of LLMs for providing accurate word descriptions and intentional disguise descriptions.

### 2.1 METHODOLOGY

The SpyGPT methodology comprises two stages: 1) prompting, in which we prompt the LLM to describe the target word using both aggressive and conservative modes, and 2) judging, where we use GPT-4 as a referee to automatically assess whether the descriptions generated by the LLM match the target word.

**Prompting** SpyGPT requires the LLM to describe a given word in two distinct modes. 1) *aggressive mode*. In the aggressive mode, the LLM is prompted to provide a clear and comprehensive description of the word {word_template} using the following prompt:

> *Please provide a focused, detailed, and accurate description of {word_template} within a limit of 100 words, so that someone can easily guess {word_template} based on the description provided.*

and 2) *conservative mode*. The LLM is instructed to provide a more ambiguous description of the word {word_template}, to accomplish disguise capability.

We employ the chain-of-thought (CoT) prompting for the LLM to perform the conservative description. First, the LLM is prompted to infer possible candidate words that are conceptually similar to the target words with the prompting:

> *Imagine other words that might share a common characteristic based on {word_template}. The candidate words may possess the same or similar attributes, and are closely related to the field of {word_template}.*

then, the LLM is instructed to generate a short description based on the common properties of generated words and the target word.

> *Please provide a conservative description of {word_template} within a limit of 10 words. You can describe the most significant commonality of these words so that others cannot guess {word_template} based on the description provided.*

Through this process, the LLM generates a description that cannot be directly inferred from the target word.

**Judging** LLMs have demonstrated significant capabilities in automatically assessing the quality of the generated text (Kocmi & Federmann, 2023; Fernandes et al., 2023). Consequently, we employ GPT-4 to evaluate the degree of correspondence between the generated descriptions and the words (the target word and the pre-defined distractor words) with the following prompts:

> *You can only reply to numbers from 1 to 5 in the following statements. Please evaluate the extent to which the description in this sentence matches the word. 1 denotes "very inaccurate" and 5 denotes "very accurate".*

**Evaluation Metrics** Target words are gathered from various sources, domains, and languages. To assess the overall performance of LLMs, we utilized two metrics: the average score on target words, and the average score on distractor words.

## 2.2 EXPERIMENT

In this study, we assess four open-source and two closed-source LLMs. The open-source models include Baichuan-7B[1], ChatGLM2-6B (Du et al., 2022), Vicuna-7B-v1.5 (Chiang et al., 2023) and Llama-2-7B-chat-hf (Touvron et al., 2023). The closed-source LLMs are GPT-3.5 (Brown et al., 2020), which utilizes Text-Davinci-002, Text-Davinci-003, and GPT-3.5-Turbo, and GPT-4, which employs GPT-4. We collect a substantial corpus of 40 target words, covering both Chinese and English languages and spanning a diverse array of fields, including social and scientific domains. We sample from the models via greedy decoding.

## 2.3 RESULT

Table 1 lists the experimental results, revealing that: 1) The closed-source GPT-4 and GPT-3.5 LLMs are significantly better than the open-source models. 2) As expected, the GPT-4 achieves the best performance in aggressive and conservative modes. Our observations are consistent with previous findings in Bubeck et al. (2023) and Qiao et al. (2023).

The advanced LLM, GPT-4, achieves a higher score of 5.00 on target words and a lower score of 1.22 for distractor words in the aggressive mode prompting. This suggests that the GPT-4 comprehends the concept associated with the target words and demonstrates the ability to describe the words accurately. On the other hand, in conservative mode prompting, the GPT-4 obtains a lower score of 4.38 for target words and a higher score of 3.06 on distractor words, indicating its ability to infer possible candidate words and its capacity to create ambiguous representations as a form of disguise.

---

[1]https://github.com/baichuan-inc/Baichuan-7B/

| Model | Aggressive Mode | | Conservative Mode | |
|---|---|---|---|---|
| | Target$_\uparrow$ | Distractor$_\downarrow$ | Target$_\downarrow$ | Distractor$_\uparrow$ |
| *Open Source Models* | | | | |
| **Baichuan-7B** | 4.08 | 1.35 | 3.27 | 1.44 |
| **ChatGLM2-6B** | 4.49 | 1.45 | 3.89 | 2.07 |
| **Vicuna-7B-v1.5** | 4.78 | 1.31 | 3.81 | 2.35 |
| **Llama-2-7B-chat-hf** | 4.78 | 1.29 | 3.89 | 2.15 |
| *Closed Source Models* | | | | |
| **Text-Davinci-002** | 5.00 | 1.28 | 4.27 | 2.49 |
| **Text-Davinci-003** | 5.00 | 1.38 | 3.68 | 2.50 |
| **GPT-3.5-Turbo** | 5.00 | 1.32 | 4.76 | 2.68 |
| **GPT-4** | 5.00 | 1.22 | 4.38 | 3.06 |
| *Human Evaluation Scores* | | | | |
| **Vicuna-7B-v1.5** | 4.79 | 2.15 | 3.83 | 2.15 |
| **GPT-3.5-Turbo** | 4.82 | 2.14 | 3.46 | 2.44 |
| **GPT-4** | 4.87 | 2.08 | 2.85 | 2.83 |

Table 1: The average scores on target words and the corresponding distractor words.

Furthermore, to address concerns regarding potential bias in using GPT-4 as an evaluation tool, we conduct a human evaluation to score the performance of various LLMs in word guessing games. The average scores of the annotators are shown in the last block of Table 1, and we include the scoring details of each annotator in Table 8 in the Appendix. The comparison validate the effectiveness of our proposed SpyGPT frameworks and ensure that the assessment results were consistent with human judgments.

# 3 AN INTERACTIVE MULTI-AGENT FRAMEWORK: SPYGAME

In this section, we first introduce the competitive language board game "Who is Spy", which is a multi-player word guessing game. Next, we describe the proposed SpyGame, an interactive multi-agent framework designed to assess the intelligence of LLMs. Finally, we present empirical results from our experiments.

## 3.1 WHO IS SPY

"Who is spy" is a strategic word game made by Happy Camp[2] in 2012. In this game, $N$ players are divided into two distinct teams: the spy team with fewer $M$ players and the villager team with the remaining $(N - M)$ players. Two conceptually similar words, e.g., "BERT" and "GPT", are distributed to players. Players cannot directly identify each other, i.e., whether they are spies or not, as they do not know the specific keywords held by others.

**Game flow** The game consists of two stages in each round: speaking and voting. In the speaking phase, players describe their keyword without revealing any characters in their keyword or deviating from it. Each player's description must be unique and not repetitive. In the voting phase, players guess the keywords of other players based on their descriptions in the speaking phase, and infer the identities of all players, including themselves. Utilizing the inferred information, they vote for a player they suspect to be the spy. The player with the most votes will be eliminated. The game continues until only the members of one team are left.

## 3.2 METHODOLOGY

Motivated by the preliminary study and "Who is Spy," we propose the interactive multi-agent framework SpyGame to evaluate the intelligence of LLMs. The SpyGame framework comprises four primary components: keyword set, host and guest agents, agent action, and victory conditions.

---

[2]https://en.wikipedia.org/wiki/Happy_Camp_(TV_series)

**Keyword Set**    To ensure the validity and fairness of the evaluation, we collect multiple keyword pairs, e.g., "BERT" and "GPT", from different sources. These keyword pairs cover various languages, topics, and domains to evaluate the LLM performance in diverse scenarios.

**Host and Guest Agents**    SpyGame utilizes several host agents (GPT-3.5-Turbo in this work) and a guest agent to participate in the game, with the guest agent assigned the role of the spy. As a participant, the guest agent remains unaware of its role as the spy, since it is not informed beforehand.

**Agent Action**    Agent action refers to the interactions among LLM-based agents. These actions are conveyed through the utterance responses generated by the LLMs. SpyGame has four distinct categories of agent actions: word guessing, speaking, reasoning, and voting. These categories facilitate effective communication and decision-making among agents.

- **Word Guessing** The agent attempts to guess another keyword based on the previous information gathered from other agents' descriptions. This requires the LLM-based agent to have a strong understanding of the context.
- **Speaking** The agent speaks based on the assigned keyword. If the agent believes it is the spy, it should describe the keyword ambiguously to hinder other players from inferring its spy identity. Otherwise, it should strategically remind its teammates of its villager identity.
- **Reasoning** In the real-world game playing scenario, human participants infer the identities of their counterparts by scrutinizing verbal and non-verbal cues, such as facial expressions and speaking tempo. Within the SpyGame framework, each LLM-based agent infers the keywords and identities of other agents based on their utterances. This necessitates a high reasoning ability of the guest agent.
- **Voting** Agents cast their votes for the agent they think is most likely to be the spy player. The agent who receives the highest number of votes is subsequently eliminated. The SpyGame's voting mechanism is performed through the LLM-based agent's responses.

**Victory Conditions**

- **Spy Victory** The guest agent, acting as the spy, successfully blends in with the host agents by generating relevant and conservative descriptions to avoid suspicion. The spy wins if it is not voted out until only two participants remain in the game.
- **Villager Victory** The host agents identify the spy by analyzing its responses and recognizing inconsistencies about their given keyword. The villagers win if they vote out the spy by a majority vote.

Due to the space limit, we list Algorithm 1 in the Appendix to illustrate the detailed process of SpyGame.

## 3.3    MODEL BIAS

Recent studies (Robinson et al., 2023; Zhao et al., 2021) have shown that LLMs exhibit an inherent selection bias in multi-choice questions. The preferences of LLMs can be influenced by the ID symbols associated with the options or the content of the prompts. Similarly, we observe the bias issue in SpyGame and identify three main bias issues, i.e., name, speaking order, and option order bias, in the SpyGame framework. To isolate the effect of variable information from the speaking phase, we test a configuration where all agents are prompted to output only a sequence of "dots" (...). The key idea is that the LLM-based agent's bias towards certain factors can be estimated using a content-free output.

**Name Bias**    LLM-based agents tend to vote for players with specific names. We design three different naming methods to evaluate the impact of name bias in SpyGame. Table 2 shows the conventional names methods. As illustrated in Figure 1, LLM-based agents tend to vote for names in positions 3 and 4 in Method 2 ("Charlie Three" and "David Four"), while showing a preference for names in positions 1 and 4 in Method 3 ("Jack" and "Tom"). Although there are slight fluctuations in Method 1, the overall variance is the smallest among the three methods. Therefore, we selected naming method 1 for our main experiments.

|            | **Name 1** | **Name 2**  | **Name 3**    | **Name 4**  |
|------------|------------|-------------|---------------|-------------|
| **Method 1** | Player 1  | Player 2    | Player 3      | Player 4    |
| **Method 2** | Aaron One | Barbara Two | Charlie Three | David Four  |
| **Method 3** | Jack      | Mary        | Alice         | Tom         |

Table 2: Examples for our three conventional naming methods.

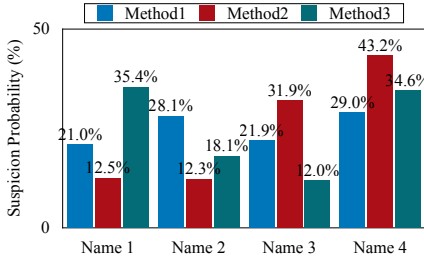

Figure 1: The suspicion probability of three naming methods.

**Speaking Order Bias**   LLM-based agents exhibit a tendency to vote for players who speak earlier in the conversation, revealing a bias towards those who first share their thoughts or opinions. The probability of being voted regarding the speaking position can be seen in Table 3. Despite the equal frequency of speakers in all positions within our permutation data-free experimental setup, we observe that agents prefer voting players in earlier positions. To mitigate this issue, we randomize the speaking order in SpyGame to ensure that agents consider all players' responses equally, regardless of their speaking position.

**Option Order Bias**   LLM-based agents tend to vote for specific options. In the voting phase, we query the agents about their suspicions and offer voting options. For instance, when inquiring about Player 1's voting preference, we use the following prompt:

> *Who do you suspect is the spy player? Player 1, it's your turn to make a choice from ['Player 2', 'Player 3', 'Player 4'] based on your previous thoughts.*

As shown in Table 3, the probability of being voted varies greatly depending on the position of the options in the given array. In our experiments, we find that the first option (Position 1) has a significantly higher probability (43.40%) of being chosen by the agents, while the second option (Position 2) holds a much lower probability (6.60%). In this experiment, each agent can only vote for the other three players, and there is no fourth position (indicated as "-"). Similar to speaking order bias, we randomize the option order in SpyGame to mitigate this issue.

In summary, addressing these biases is crucial for ensuring a fair and accurate evaluation of the LLM-based agents' intelligence in the SpyGame framework. By randomizing speaking and option orders, and using a diverse set of names, we can effectively mitigate these biases and improve the overall fairness and validity of the evaluation process.

## 3.4 EXPERIEMNT

**Setup**   We establish a four-player setting for SpyGame in which three host agents are consistently designated as GPT-3.5-Turbo LLMs. Subsequently, we assess different LLMs by assigning them the role of the spy. For the keyword set, we gather 50 pairs (50 x 2 = 100) of keywords. For each LLM under evaluation, we conduct 100 experiments for each keyword allocated to the LLM.

| **Bias Type** | **Position 1** | **Position 2** | **Position 3** | **Position 4** |
|---------------|----------------|----------------|----------------|----------------|
| **Speaking**  | 32.47          | 27.60          | 20.31          | 19.62          |
| **Option**    | 43.40          | 6.60           | 50.00          | -              |

Table 3: The probability of being voted regarding speaking and option order bias.

**Evaluation Metrics**   We define three metrics to evaluate the performance of the guest agent LLMs: 1) **Win** represents the average win rate of the guest agent in 100 games. 2) **Round** indicates the average number of rounds the guest agent survives. 3) **Voted** refers to the average number of votes the guest agent receives per round.

## 3.5   REUSLT

| Method | Spy | | |
|---|---|---|---|
| | Win↑ | Round↑ | Voted↓ |
| Text-Davinci-002 | 0.16 | 1.99 | 1.49 |
| Text-Davinci-003 | 0.18 | 2.03 | 1.40 |
| GPT-3.5-Turbo | 0.21 | 2.04 | 1.47 |
| GPT-4 | 0.33 | 2.18 | 1.31 |

Table 4: The performance of different LLM-based guest agents in SpyGame.

Table 4 presents the results of the SpyGame experiments. GPT-4 outperforms other models in terms of all three metrics, indicating its superior ability to deceive host agents and avoid suspicion as the spy. Meanwhile, the performance of the Text-Davinci series models is consistent with the single-round SpyGPT results shown in Table 1. These models are less effective in generating relevant and ambiguous responses to conceal their spy identity. Our experiment showcases the potential of using SpyGame as a framework for evaluating LLM-based agents' intelligence and reasoning capabilities in a competitive and interactive setting.

## 4   ANALYSIS

### 4.1   ABLATION STUDY

| Method | Spy | | |
|---|---|---|---|
| | Win↑ | Round↑ | Voted↓ |
| GPT-4 | 0.33 | 2.18 | 1.31 |
| w/o Word Guessing | 0.26 | 2.12 | 1.34 |
| w/o Reasoning | 0.21 | 2.08 | 1.40 |

Table 5: Ablation study on the impact of word guessing and reasoning actions in the SpyGame for the guest agent GPT-4.

In the ablation study, we analyze the impact of word guessing and reasoning actions as described in Section 3.2. The results of the ablation study are shown in Table 5. We can observe that the performance of the guest agent GPT-4 without word guessing drops in terms of *Win* (from 0.33 to 0.26) and *Round* (from 2.18 to 2.12). Without word guessing action, the guest agent is not aware of other people's keyword and is more likely to speak aggressively, which could potentially reveal its spy identity. Similarly, when the reasoning action is removed, the overall performance significantly declines in all three metrics. Reasoning reflects the agent's ability to infer the other players' identities and is beneficial for making better decisions in the next round.

### 4.2   ROBUSTNESS

To evaluate the robustness of the SpyGame, we perform the SpyGame experiment with GPT-4 as the guest agent and conduct another two group experiments in the same experimental settings (i.e., three GPT-3.5-Turbo as host agents). Considering the bias issues mentioned in Section 3.3, we utilize different random seeds in each series to ensure that the order of agent responses changes.

As shown in Table 6, SpyGame achieves stable performance in both victory rate and the number of survival rounds. Although there is variance among the more fine-grained *Voted* metric, all sets of

GPT-4 outperform other LLMs consistently (refer to Table 4). This demonstrates the effectiveness of SpyGame in providing a reliable result of different models.

| Method | Spy | | |
|---|---|---|---|
| | Win$_\uparrow$ | Round$_\uparrow$ | Voted$_\downarrow$ |
| GPT-4 | 0.33 | 2.18 | 1.31 |
| GPT-4 / seed-1 | 0.34 | 2.19 | 1.22 |
| GPT-4 / seed-2 | 0.32 | 2.18 | 1.28 |

Table 6: Robustness experiment of the SpyGame with GPT-4 with different random seeds.

### 4.3 THEORY OF MIND

Spy agents must accurately infer their spy role and win the game by concealing their true intentions, which is challenging even for human players. The entire process involves one crucial ability that we are particularly interested in the reasoning ability to deduce the identities of all the participants. As pointed out by Bubeck et al. (2023), the reasoning ability mentioned in this work has a more precise and clear definition in psychology, known as the Theory of Mind (ToM).

Cognitive ToM is divided into first-order, which involves reflecting on someone else's mental states, and second-order, which involves reflecting on someone's perception of another person's mental state Doody et al. (1998). In this context, we define the first-order ToM as understanding another person's thoughts (e.g., What is Player 1's keyword?), and the second-order ToM as inferring what one person thinks about another person's thoughts (e.g., What is the identity does Player 1 guesses for Player 2?)

**Setup** We conduct the ToM analysis using the game history log in the main experiments (Section 3.4). Specifically, we analyzed the historical memory prior to the first round of voting, as all players had already made their first speaking and no players had been eliminated at that point.

**First-Order ToM** For the first-order, we gain insight into the guest's inference regarding the keywords and identities of others with the prompting:

> *It is your turn to guess the keywords and identities of all players, including yourself.*
> *You must identify only one spy player.*

Through the reasoning strategy, We define the inference of others' keyword as *1-word* metric, the inference of others' identity as *1-identity*. In addition, we also prompt the guest agent to infer its own identity, referred to as *self-identity*.

**Second-Order ToM** For the second-order ToM, we prompt the guest agent with the following instructions:

> *Based on your description, what do you think other players will guess your keyword and identity to be? Please put yourself in the shoes of other players and guess your own keyword and identity.*

| Method | Theory of Mind | | | | |
|---|---|---|---|---|---|
| | Self-Identity | 1-Word | 1-Identity | 2-Word | 2-Identity |
| Text-Davinci-002 | 0.20 | 0.22 | 0.72 | 0.27 | 0.47 |
| Text-Davinci-003 | 0.14 | 0.25 | 0.72 | 0.35 | 0.61 |
| GPT-3.5-Turbo | 0.23 | 0.17 | 0.77 | 0.37 | 0.59 |
| GPT-4 | 0.17 | 0.38 | 0.72 | 0.35 | 0.67 |

Table 7: Theory of mind performance of guest agents.

We use the first-order inference of other host agents as ground truth, aiming to evaluate the target LLM's ability to infer the thoughts of other agents accurately.

As is shown in Table 7, the performance of different models varies across different ToM metrics. For *self-identity*, GPT-3.5-Turbo performs the best with a score of 0.23. In terms of first-order ToM, GPT-4 has the highest score in *1-word* with 0.38, while GPT-3.5-Turbo leads in *1-identity* with a score of 0.77. For second-order ToM, GPT-4 also performs well in both *2-word* and *2-identity* metrics, with scores of 0.35 and 0.67. These results indicate that LLM-based agents have varying levels of success in understanding and attributing mental states to themselves and others.

## 5    RELATED WORK

**Evaluation of LLMs**    The evaluation of LLMs has become an essential area of research, covering three primary categories. Firstly, NLP tasks involve diverse applications aimed at understanding and generating textual data. Prominent benchmarks in this category include include GLUE (Wang et al., 2018), SuperGLUE (Wang et al., 2019) MMLU (Hendrycks et al., 2020). Secondly, alignment evaluation assesses the helpfulness and harmlessness of LLM-generated text (Bai et al., 2022)with examples such as instruction-following assessments like AlpacaEval (Li et al., 2023b). Lastly, the third category involves complex real-world tasks, as exemplified by Webshop (Yao et al., 2022), AgentBench (Liu et al., 2023), Webarena (Zhou et al., 2023), which test LLMs ability to handle intricate and practical scenarios. As discussed in Section 1, creating these human-annotated benchmarks can be time-consuming and costly, as it requires domain expertise and extensive manual labor. More critically, this category of methods is plagued by data leakage issues (Schaeffer, 2023).

**LLM-based Agent**    More recently, the LLM-based agent has drawn significant attention with the rapid development of LLMs. In the field of NLP, communicative agents that leverage the power of LLMs to generate coherent responses and engage in multi-turn conversations, simulating human-like communication patterns, have been proposed to improve the reasoning and factuality in natural language generation (Du et al., 2023; Liang et al., 2023). The communicative agents can also be applied across a wide range of real-world applications, including software development (Qian et al., 2023; Hong et al., 2023), social simulation (Park et al., 2022; 2023) and robot assistance (Brohan et al., 2023; Wu et al., 2023).

**Game Playing with LLMs**    Several recent studies have attempted to incorporate LLMs into games, e.g., GameEval (Qiao et al., 2023), and Werewolf (Xu et al., 2023). These efforts aim to explore the potential of LLMs in-game settings, examining their adaptability, strategic thinking, and ability to engage in complex interactions with other players. The core differences between our work and GameEval are two-fold: 1) the objective of our work is to evaluate the expression and disguising abilities of LLMs, and 2) we observe biases in the multi-agent interaction framework and propose a more comprehensive evaluation framework to address the issue.

## 6    CONCLUSION

In this paper, we propose employing the word guessing game to assess the LLM-based agent intelligence automatically. To this end, we develop a single-agent assessment method, SpyGPT, and an interactive multi-agent framework, SpyGame. The proposed SpyGPT and SpyGame can be easily migrated to various tasks, domains and languages. SpyGPT requires the target LLM to describe the given word in the aggressive and conservative modes and utilizes the SOTA GPT-4 to determine whether the descriptive sentences can accurately describe the given word. Empirical experimental results and human evaluation demonstrate that the SpyGPT can effectively evaluate the intelligence of LLMs. SpyGame leverages the agent competition to facilitate the exploration of LLMs' expressive language abilities and their theory of mind intelligence in intricate communication contexts. We identify three primary bias issues in multi-agent gameplay experiments and propose a simple and effective strategy for mitigating these biases. Extensive experiments and analysis demonstrate that the proposed SpyGame can effectively evaluate the capabilities of various LLMs in multi-agent interaction, capturing their ability to adapt to novel situations and engage in strategic communication.

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

## A   APPENDIX

| Model | Aggressive Mode | | Conservative Mode | |
|---|---|---|---|---|
| | Target$_\uparrow$ | Distractor $_\downarrow$ | Target$_\downarrow$ | Distractor$_\uparrow$ |
| | Human Annotator 1 | | | |
| **Vicuna-7B-v1.5** | 4.92 | 2.11 | 3.24 | 1.89 |
| **GPT-3.5-Turbo** | 4.95 | 2.11 | 3.08 | 2.14 |
| **GPT-4** | 5.00 | 2.08 | 2.08 | 2.38 |
| | Human Annotator 2 | | | |
| **Vicuna-7B-v1.5** | 4.65 | 2.19 | 4.41 | 2.41 |
| **GPT-3.5-Turbo** | 4.68 | 2.16 | 3.84 | 2.73 |
| **GPT-4** | 4.73 | 2.08 | 3.62 | 3.27 |

Table 8: The Human Evaluation scores on target words and the corresponding distractor words.

---

**Algorithm 1** SpyGPT: Interactive Multi-Agent Framwork

---

**Require:** Keyword pair $\{i, j\}$, number of all agents $N$ and guest agent $X$
**Ensure:** Final winning team $t$

1: **procedure** SPYGPT($\{i, j\}$, $N$, $X$)
2:     $W_{spy} \leftarrow Random\ Selection(i, j)$                            ▷ Initialize spy team's keyword
3:     $W_{villager} \leftarrow Select\ w \in \{i, j\} \cap w \neq W_{spy}$          ▷ Initialize villager team's keyword
4:     $H_{spy} \leftarrow [W_{spy}, N]; H_{villager} \leftarrow [W_{villager}, N]$          ▷ Initialize game history
5:     $X \leftarrow [H_{spy}]; Y_1, \cdots, Y_{N-1} \leftarrow [H_{villager}]$          ▷ Initialize agents
6:     $P \leftarrow [X, Y_1, \cdots, Y_{N-1}]; N_{survive} \leftarrow N$          ▷ Record all agents
7:     **while** $N_{survive} > 2$ **do**
8:         **for** each $P_i$ in $P$ **do**                                        ▷ Speaking phase
9:             $s \leftarrow P_i(H)$                                        ▷ Generate desciptions
10:            $H \leftarrow H + [s]$                                        ▷ Append $s$ to $H$
11:        $V \leftarrow []$                                        ▷ Initialize number of votes
12:        **for** each $P_i$ in $P$ **do**                                        ▷ Voting phase
13:            $v \leftarrow P_i(H)$                                        ▷ Generate voted agent
14:            $H \leftarrow H + [v]$                                        ▷ Append $v$ to $H$
15:            $V \leftarrow V + [v]$                                        ▷ Append $v$ to $V$
16:        $P_{voted} \leftarrow Max\ p \in V$                                        ▷ Select the voted agent
17:        **if** $P_{voted} = X$ **then**
18:            break                                        ▷ Spy agent out, game over
19:        **else**
20:            $P \leftarrow P - P_{voted}; N_{survive} \leftarrow N_{survive} - 1$          ▷ Villager agent out, game continue
21:     **return** $t$

---

