# OpenReview forum: "Leveraging Word Guessing Games to Assess the Intelligence of Large Language Models"
_ICLR.cc/2024/Conference — ICLR 2024 Conference Withdrawn Submission_

### Official Review · Reviewer_BUuv · 2023-10-30

**Soundness:** 2 fair
**Presentation:** 3 good
**Contribution:** 1 poor
**Rating:** 3
**Confidence:** 3

**Summary:**

This paper addresses the need for automatic evaluation methods for Language Model (LLM)-based agents. The authors propose an evaluation framework inspired by language games like "Who is Spy" and "SpyFall," involving a word guessing game . They introduce SpyGPT to evaluate LLMs' descriptive and disguising abilities and SpyGame, an interactive multi-agent board game to assess their linguistic skills and strategic thinking. Experiments with this framework demonstrate its effectiveness in evaluating LLMs' adaptability and strategic communication capabilities in complex scenarios.

**Strengths:**

Clarity
- The proposed methodology is clearly explained and easy to follow.

Quality:
- The paper provide a good amount of LLM, both open and "closed" source LLMs, and provide human evaluation to verify the effectiveness of the proposed method.

**Weaknesses:**

Originality
- using text based games, both interactive and statics has been widely explore in the research community to evaluate LLMs.

Significance
- overall, I cannot see any evidence that this metric 1) strongly correlated with human judgment of how good the LLMs in more realist and meaningful scenarios, 2) how to rank models based on the proposed metric.

**Questions:**

Nan

---

### Official Review · Reviewer_hXPk · 2023-10-31

**Soundness:** 2 fair
**Presentation:** 3 good
**Contribution:** 2 fair
**Rating:** 3
**Confidence:** 5

**Summary:**

This paper introduces a single-agent framework SpyGPT and a multi-agent framework SpyGame, based on the board game Who Is Spy. The authors convert the board game to a multi-agent LLM setup, where all agents are motivated to apply reasoning before voting who the spy is. Specifically, given a target word and a distracting word, the host agents are prompted to identify the guest agent, and the guest agent is prompted to hide and misguide other agents through strategies such as interpreting other agents' roles. Results show that GPT-4 achieves the best performance. Furthremore, the authors identify and ablate three bias issues including name, speaking order, and option order.

**Strengths:**

1. This paper introduces an interesting multi-agent evaluation framework SpyGame, which requires strong reasoning capabilities.
2. What is more challenging about this environment is that in order to win the game, an agent needs to interpret other agents' reasoning, i.e., Second-Order Theory Of Mind. This can be a good benchmark for LLM evaluations, especially for reasoning and multi-agent.
3. The paper is generally well written.

**Weaknesses:**

1. Although the paper is well written, it is not easy to actually reproduce the experiments. Most importantly, given that prompting is critical to the evaluation in this paper, there is no detail in terms of what prompts are actually used, and if few-shot examples are applied to evaluate either the SpyGame or SpyGPT (Also I suggest using a different name because my understanding is that you did not train a GPT model specially for the SpyGame).
2. Similar, there is no information in terms of what "40 target words" are used, and how "a sequence of dots" are used to remove bias in the model. For the bias part, there is no detail regarding how many games are sampled to show how sever the bias problem is.
3. Rather than changing orders to remove bias, it seems that a better solution is to use random tokens (rather than "player 1"), which does not require shuffling, and may not lead to any of the bias problem studied.

**Questions:**

1. Have you tried using other agents (other than GPT3.5) as the host agents for evaluation (e.g., use GPT-4 for all agents)?
2. What is "word_template" in the prompt? How is this different from a "word" or a "word description"?

---

### Official Review · Reviewer_c2gN · 2023-10-31

**Soundness:** 3 good
**Presentation:** 4 excellent
**Contribution:** 2 fair
**Rating:** 5
**Confidence:** 4

**Summary:**

The paper proposes a method for assessing language models by leveraging word guessing games, specifically the SpyFall game. The authors tested the ability of LLMs to generate world descriptions and intentional disguise descriptions, and introduced the framework of the SpyGame, which includes agent actions and voting. The authors also conducted model bias evaluations to ensure fairness and accuracy in the evaluation process.

**Strengths:**

1. The idea of using games to assess LLMs is interesting and unique, as it introduces multiple agents and an active environment.
2. The paper is well-written and easy to understand. The authors carefully designed the game, and the model bias evaluation adds insight and fairness to the evaluation process.

**Weaknesses:**

1. The paper only focuses on the SpyFall game, and it is not clear why this game is important for the research community or real-world applications. It would be beneficial to carefully consider other potential games and select a representative subset of them.
2. The method for prompting LLMs to play the SpyGame is not described in detail in section 3 or the appendix.

**Questions:**

1. Why was SpyFall chosen as the game for the evaluation, and what unique properties does it have compared to other language games?
2. How did the authors handle the potential for mistakes from the host agents, and what measures were taken to ensure accuracy in the evaluation process?
3. In the robustness study, how does the seed affect the results, and how was the robustness of GPT-4 and GPT-3.5-turbo evaluated as hosts?
4. In Table 1, why are the results of the model evaluation and human evaluation seemingly reversed? How do the evaluation results align with human evaluation?